# Effects of a low-carbohydrate diet in adults with type 1 diabetes management: A single arm non-randomised clinical trial

Jessica L. Turton[1]*, Grant D. Brinkworth[2], Helen M. Parker[1], David Lim[3], Kevin Lee[4], Amy Rush[5], Rebecca Johnson[5], Kieron B. Rooney[1]

1 Faculty of Medicine and Health, University of Sydney, Camperdown, New South Wales, Australia, 2 CSIRO – Health and Biosecurity, Westmead, New South Wales, Australia, 3 Church Street Medical Practice, Newtown, New South Wales, Australia, 4 Qscan Group, Clayfield, Queensland, Australia, 5 Type 1 Diabetes Family Centre, Stirling, Western Australia, Australia

* Jessica.turton@sydney.edu.au

**Data Availability Statement:** All relevant data are within the paper and its Supporting information files.

## Abstract

Public interest in low-carbohydrate (LC) diets for type 1 diabetes (T1D) management has increased. This study compared the effects of a healthcare professional delivered LC diet compared to habitual diets higher in carbohydrates on clinical outcomes in adults with T1D. Twenty adults (18–70 yrs) with T1D ($\geq$6 months duration) with suboptimal glycaemic control (HbA1c>7.0% or >53 mmol/mol) participated in a 16-week single arm within-participant, controlled intervention study involving a 4-week control period following their habitual diets (>150 g/day of carbohydrates) and a 12-week intervention period following a LC diet (25–75 g/day of carbohydrates) delivered remotely by a registered dietitian. Glycated haemoglobin (HbA1c –primary outcome), time in range (blood glucose: 3.5–10.0 mmol/L), frequency of hypoglycaemia (<3.5 mmol/L), total daily insulin, and quality of life were assessed before and after the control and intervention periods. Sixteen participants completed the study. During the intervention period, there were reductions in total dietary carbohydrate intake (214 to 63 g/day; P<0.001), HbA1c (7.7 to 7.1% or 61 to 54 mmol/mol; P = 0.003) and total daily insulin use (65 to 49 U/day; P<0.001), increased time spent in range (59 to 74%; P<0.001), and improved quality of life (P = 0.015), with no significant changes observed during the control period. Frequency of hypoglycaemia episodes did not differ across timepoints, and no episodes of ketoacidosis or other adverse events were reported during the intervention period. These preliminary findings suggest that a professionally supported LC diet may lead to improvements in markers of blood glucose control and quality of life with reduced exogenous insulin requirements and no evidence of increased hypoglycaemia or ketoacidosis risk in adults with T1D. Given the potential benefits of this intervention, larger, longer-term randomised controlled trials are warranted to confirm these findings.

**Trial Registration**: https://www.anzctr.org.au/ACTRN12621000764831.aspx

**Funding:** This study has been partially funded by The Commonwealth Scientific and Industrial Research Organisation (CSIRO) who kindly donated $6000 AUD to this research project. We also raised philanthropic funds (University of Sydney crowd-sourcing platform) to the amount of ~$7000 AUD to cover remaining costs. The Type 1 Diabetes Family Centre agreed to reimburse the study diabetes educator for their time (up to 30 hours) so they could perform their necessary role in this research. The funders had no role in study design, data collection and analysis, decision to publish, or preparation of the manuscript.

**Competing interests:** I have read the journal's policy and the authors of this manuscript have the following competing interests: All healthcare practitioners involved in the conduct of the study (Jessica Turton, Kevin Lee, David Lim, and Amy Rush) have private consulting businesses in Australia. The outcome(s) of the intervention may be considered a reflection of their proficiency as healthcare practitioners. Nevertheless, existing patients of Ms Turton, Dr Lee, Dr Lim, and Mrs Rush were excluded from participating in the study. This does not alter our adherence to PLOS ONE policies on sharing data and materials.

## Introduction

Type 1 diabetes (T1D) is an autoimmune condition characterised by pancreatic beta cell destruction, absolute insulin deficiency, and impaired glucose metabolism [1]. Despite modern advancements in glucose monitoring and insulin delivery technologies, many individuals with T1D experience high variability in blood glucose levels and difficulties achieving glycaemic targets (glycated haemoglobin [HbA1c] ≤7.0% or ≤53 mmol/mol) [2–6], increasing the risk for several acute and chronic health complications including cardiovascular disease (CVD) [1, 3, 7–13]. Consequently, effective treatment strategies that achieve glycated haemoglobin targets while minimising the frequency and severity of hyper- and hypoglycaemia are needed.

Dietary carbohydrates significantly influence post-prandial blood glucose levels and individuals with T1D are recommended to quantify carbohydrate intake (known as 'carbohydrate counting') to predict rises in post-prandial glucose levels and administer appropriate insulin dosages for meals and snacks [3]. Authoritative bodies including the National Health & Medical Research Council (NHMRC), recommend individuals with T1D follow the National Dietary Guidelines which promote a high-carbohydrate diet (HC; 45–65% of total energy intake [TEI]) [3, 14]. A recent analysis of the Australian Health Survey showed that dietary intakes of adults with T1D were consistent with these recommendations [15]. However, data from the Australian National Diabetes Audit report that average HbA1c levels remain at 8.4% (68 mmol/mol) in this clinical population [16], suggesting that alternative dietary approaches could be considered to improve glycaemic control.

In adults with type 2 diabetes (T2D), randomised controlled trials have repeatedly demonstrated that low-carbohydrate (LC) diets (≤130 g/day or 26% TEI from carbohydrates) achieve greater reductions in HbA1c and anti-glycaemic medications, with greater increases in HDL-cholesterol and decreases in triglycerides, when compared to traditional HC diets [17–19]. Conversely, few high quality studies have investigated the role of LC diets in adults with T1D [20]. In 2018, the first published systematic review examining the effects of lower-carbohydrate diets (<45% TEI) for T1D management reported that the three included studies examining LC diets (<26% TEI; excluding case reports) achieved mean HbA1c reductions between 0.7–1.3% [20–23], and diets with ≤100 g/day of carbohydrates led to concurrent reductions in total daily insulin use [21–23]. Further, a 2019 randomised cross-over trial conducted in 10 adults with T1D showed that a 12-week LC diet (~100 g/day) resulted in less time spent with blood glucose levels below 3.9 mmol/L and lower glycaemic variability compared to a HC diet (~250 g/day) [24]. A recent retrospective analysis of adults with T1D who self-selected to follow a professionally supported LC diet (~60 g/day of carbohydrates) reported reductions in HbA1c, fasting blood glucose levels, and total daily insulin use, with increased time spent in target glucose ranges [25]. Despite preliminary evidence suggesting effectiveness of LC diets for T1D management, lack of consensus from authoritative bodies and healthcare professionals regarding the use and feasibility of LC diets for patients with T1D remains [26–28], and further prospective interventional studies are needed.

The objective of this study was to compare the effects of a professionally supported LC diet intervention with habitual diets higher in carbohydrates on clinical markers including HbA1c, glycaemic variability, frequency of hypoglycaemia, total daily insulin (TDI), and quality of life in adults with T1D. It was hypothesised that that a LC diet intervention would achieve greater improvements in clinical markers of T1D management.

## Materials and methods

### Trial design

In this 16-week single arm within-participant controlled intervention study, adults with T1D completed a 4-week control period followed by a 12-week intervention period consisting of a dietitian-delivered LC diet (25–75 g/day digestible carbohydrates). Since multiple lifestyle factors influence T1D management, and with consideration of individual personal needs and preferences, a single arm within-patient intervention study where participants act as their own controls was deemed to be the most appropriate design. The primary outcome was HbA1c. Secondary outcomes were glycaemic variability (GV), frequency of hypoglycaemia, total daily insulin use, and quality of life. The study protocol containing full details of this clinical trial has been published elsewhere [29]. While originally intended to be delivered in-person, the trial commenced during COVID-19 lockdowns (Sydney, July 2021) and was completed remotely in its entirety, using Telehealth and remote testing services that permitted Australia-wide participation. The most recent version of the study protocol reflecting these updates is available as supporting information, alongside the TREND checklist for non-randomised trials (S1 File, S1 Table). Data collection commenced in July 2021 and was completed in July 2022. This trial was reviewed and approved by the University of Sydney Human Research Ethics Committee (project number: 2021/080) and prospectively registered with the Australian New Zealand Clinical Trials Registry (ANZCTR) (https://www.anzctr.org.au/ACTRN12621000764831.aspx). All participants provided written informed consent.

### Participants

Participants were recruited via public advertisement, including posters/flyers displayed on social media from July 2021 to March 2022. Eligible participants were aged 18–70 years with a body mass index (BMI) between 18.5–39.9 kg/m$^2$, confirmed diagnosis of T1D ($\geq$6 months), using multiple daily insulin injections or an insulin pump, and an HbA1c >7.0% (>53 mmol/mol). Participants were required to provide written evidence of their HbA1c result (e.g., pathology report, specialist letter) measured within three months of screening, and were not excluded if their HbA1c reduced $\leq$7.0% ($\leq$53 mmol/mol) at the time of study commencement. Participants had to reside within Australia for the duration of the trial and have a habitual intake of digestible carbohydrates >150 g/day. Exclusion criteria included: non-English speaking; habitual use of an automated insulin delivery system and/or adherence to a fixed insulin regimen; previously diagnosed hypo-unawareness; habitual dietary intake strictly excluding animal-based proteins (e.g., vegan diet); recent pregnancy or lactation ($\leq$6 months); self-identifies as current or recent smoker ($\leq$6 months); recent weight change >10% body weight ($\leq$3 months); a known family history of heart disease; previously diagnosed with familial hypercholesterolaemia, gastrointestinal disease (not including irritable bowel syndrome, coeliac disease or stable inflammatory bowel disease), liver disease (not including fatty liver), chronic kidney disease (eGFR <60 mL/min/1.73m$^2$), respiratory disease (not including stable treated asthma), thyroid disease (not including stable treated hyper- or hypothyroidism) or CVD; or, previously diagnosed with an eating disorder. Patients of the study investigators were also excluded. Prior to study commencement, participants were required to nominate and confirm a member of their usual diabetes care team proficient in insulin management (endocrinologist, general practitioner, or diabetes educator) to provide ongoing support throughout the study.

## Control period

Details of the control period have been reported elsewhere [29]. In brief, the study dietitian (J. L.T.) instructed participants to maintain habitual eating, exercise and T1D management throughout the initial 4-week control period. Participants were provided with standard diabetes education and instructed to test their blood glucose levels six times daily for self-monitoring purposes [3, 30].

## Diet intervention

Details of the dietary intervention are reported elsewhere [29]. In brief, the study dietitian met with participants individually via Zoom video conferencing (Zoom) on six occasions (60 min each) throughout the 12-week diet intervention period to provide instruction, education, and strategies to follow a LC diet. The LC diet prescription was informed by a published systematic review assessing previously reported LC diet interventions shown to be safe and effective for improving glycaemic control in adults with T2D [31]. The carbohydrate prescription started at 50 g of digestible carbohydrates per day, with opportunity to be adapted within a broader range of 25–75 g/day according to individual blood glucose levels and personal preference. Participants were encouraged to distribute carbohydrates evenly across the day such that total carbohydrates did not exceed 20 g at a single eating occasion. Dietary education incorporated information on the post-prandial effect(s) of carbohydrates, proteins and fats, including carbohydrate and protein counting, given the need to consider protein in calculating meal-time insulin requirements with habitual carbohydrate restriction <100 g/day [32]. Participants were provided with an educational booklet containing sample meal plans, troubleshooting tips, and ideas for meals and snacks. Examples of the meal plans for 25 g/day, 50 g/day and 60 g/day of dietary carbohydrates are provided as S2–S4 Tables. Consumption of whole foods was emphasised and a food list showing options for proteins, fats, and carbohydrates was provided. No upper intake limit on recommended sources of proteins and fats was provided. Alcohol intake recommendations for the general population were provided [33]. Participants were encouraged to maintain their usual physical activity level throughout the study duration.

Participants were instructed to perform usual care practice self-monitoring blood glucose readings before and two hours after each meal (measured using their own blood glucose monitoring device) for insulin calculations and adjustments throughout the study. Blood glucose targets were consistent with standard practice for diabetes management (4–8 mmol/L when fasting and before meals, and 4–10 mmol/L two hours after starting meals) [3, 30]. Participants were instructed to measure blood ketones at least twice per week, and more frequently if feeling unwell, and were to maintain ketones ≤0.6 mmol/L by following their sick day management plan, as per standard diabetes practices [3, 34].

Participants were provided an information booklet about insulin management on a LC diet and received access to short, pre-recorded videos (3–5 minutes each) by the study diabetes educator (A.R.) explaining this information. Participants were provided the opportunity to meet with the study diabetes educator via Zoom for at least one 30-minute session during the intervention period to discuss questions relating to diabetes management on a LC diet. Participants were provided fortnightly reminders to follow up with their nominated healthcare practitioner for individualised advice on insulin titrations.

The participants' other medications such as oral anti-glycaemic and anti-hypertensive agents were assessed prior to commencing the intervention by the study physician (D.L.) to develop a medication management plan that informed participants' usual GP of the expected adjustments that may be required with adherence to a LC diet.

## Outcome measures

Outcomes were measured at three time-points: (1) pre-control period (-4 weeks), (2) post-control (0 weeks), and (3) post-intervention period (12 weeks). HbA1c, fasting blood glucose, kidney function (sodium, potassium, chloride, bicarbonate, urea, creatinine, eGFR, calcium, corrected calcium, phosphate, uric acid), liver function (total protein, albumin, alkaline phosphatase, total bilirubin, GGT, AST, ALT, globulin, magnesium, creatine kinase), and lipid studies (total cholesterol, low density lipoprotein (LDL) cholesterol, high density lipoprotein (HDL) cholesterol, triglycerides) were assessed from fasting blood samples collected and analysed by locally available NATA-accredited laboratories using standard procedures. Participants were advised to fast (water as required) for 10–12 hours and avoid strenuous exercise 24-hours prior to blood testing.

Continuous glucose monitoring (CGM) devices were provided to participants (Medtronic Australasia iPro2 or Guardian Connect, depending on availability) to obtain 24-hr interstitial glucose concentrations for seven days at each data collection timepoint. Participants unwilling or unable to use the CGM provided used their own commercially available device. Raw data were entered into the EasyGV platform (EasyGV Version 9.0.R2, University of Oxford, Oxford, UK) for calculation of glycaemic variability (GV) indices including standard deviation of blood glucose (SDBG), mean amplitude of glycaemic excursions (MAGE), and mean blood glucose. Time in range (TIR) was calculated by one researcher (J.L.T.) to identify the percentage of datapoints within 3.5–10.0 mmol/L across the entire data collection period. Frequency of hypoglycaemia was defined as the number of events <3.5 mmol/L with or without symptoms. GV data were analysed with and excluding days that had incomplete values (<80% available datapoints), but no meaningful differences were identified. Data without exclusions (i.e., all CGM data regardless of completeness) were used for the primary analysis; analysis of data with exclusions are provided in the S5 Table.

Total daily insulin (TDI) was defined as the sum of all basal and bolus insulin given over a 24-hour period derived from a 3-day self-report insulin log. Participants using insulin pumps were provided the option to provide pump summary reports instead of completing the insulin log.

Anthropometric outcomes, including BMI, waist circumference, and resting blood pressure were measured by a local healthcare professional (i.e., local pharmacist, GP, nurse). In cases where participants were unable or unwilling to leave their homes during COVID-19 lockdowns, self-reported measures were provided.

Quality of life was assessed using the Diabetes-related Quality of Life (DQoL) Brief Clinical Inventory [35] completed online. The 15-item questionnaire is a quantitative assessment for the perceptions of how diabetes mellitus affects daily function [35, 36]. Items are ranked on a 5-point Likert scale (1 = "very satisfied" or "never" up to 5 = "very unsatisfied" or "constantly"), with the sum of all 15 items providing a total score from 15 to 75. A lower score implies a more satisfactory quality of life.

Participants completed 3-day weighed food records using the smartphone app, *Easy Diet Diary* (Xyris, version 6.0.28, Australia), at each data collection timepoint. Data was analysed by the study dietitian (J.L.T.) using *FoodWorks Professional Edition* (Xyris, version 10, Australia) to assess dietary intake (total energy, total digestible carbohydrate, dietary fibre, protein, total fat, saturated fat, and alcohol). Diet satisfaction was assessed using a 6-item questionnaire previously used in T2D research [37], and total physical activity level (PAL) was measured using the International Physical Activity Questionnaire (IPAQ) [38].

## Statistical analyses

The primary outcome was the change(s) in HbA1c across timepoints (pre-control, post-control, and post-intervention). Based on a clinically relevant difference in HbA1c of 0.7% (absolute), with a standard deviation of 1.0, to achieve 80% power with alpha <0.05, a sample size of n = 16 was determined using a paired t-test of comparisons [22]. The primary analysis was conducted on participants with complete data only, and secondary analysis was conducted on an intention-to-treat (ITT) basis with the last recorded measurement carried forward for missing values. Prior to analysis, data was assessed for normality by two investigators (K.B.R. and J.L.T.) using histograms and PP-plots. Parametric and nonparametric data were assessed using repeated measures ANOVA and the Friedman test, respectively, to determine within-group differences across timepoints. If a significant main effect was identified (p<0.05), post hoc analysis using paired t-tests (parametric) or Wilcoxon signed-rank tests (non-parametric) was performed to compare two pairs (pre-control vs. post-control and post-control vs. post-intervention) with a Bonferroni correction applied, resulting in a significance level set at p<0.025. Data are presented as means and standard deviations (SD) (normally distributed data) or medians and interquartile ranges (IQR) (non-normally distributed data) for each timepoint. Statistical analyses were conducted using *SPSS Statistics* (released 2021, IBM SPSS Statistics for Windows, version 28; Armonk, New York).

## Results

Twenty participants commenced, and 16 participants completed the intervention with post-intervention data collected (Fig 1). Reasons for drop-out included pre-existing mental health issues and experienced difficulties meeting the study requirements (n = 1), difficulty achieving insulin adjustment requirements (n = 1), a lack of time to meet the study requirements (n = 1), and loss to follow up (n = 1). Both completers (n = 16) and ITT analyses (n = 20) showed a similar pattern of results. Results for the completers analysis are presented here with the ITT results presented in S6–S9 Tables. Of the 16 completers, the CGM devices used to collect 7-day continuous blood glucose data for assessing GV outcomes included ipro2 CGM device (Medtronic Australasia) (n = 8), the Guardian Connect (Medtronic Australasia) (n = 8), FreeStyle LibreLink (Abbott, Australia) (n = 2), and Dexcom G6 (Dexcom Inc., United States) (n = 1).

### Baseline characteristics

At baseline, mean age of participants was 43 years, 50% were female, and mean duration of T1D was 21 years (Table 1). Eleven participants (69%) were habitually using a CGM to monitor their glucose levels and 50% were using an insulin pump (Table 1). Nine participants (56%) had one or more medical conditions other than diabetes and 11 participants (69%) were taking medications other than insulin, with the most common being cholesterol-lowering medications (Table 1, S10 Table).

### Dietary intake

Dietary intake data are provided in Table 2 (n = 16). Total carbohydrate intake and percent of total energy intake (%TEI) from carbohydrates reduced by 151 ± 65 g/day and 23 ± 9%, respectively from post-control to post-intervention (P<0.001), with no significant difference between pre- and post-control. Absolute total protein intake did not significantly change across the study, but the proportion of TEI (%TEI) from protein increased by 7 ± 4% from post-control to post-intervention (P<0.001), with no significant difference between pre- and post-control. Total dietary fat intake and %TEI from dietary fats increased by 34.3 ± 41.3 g/day

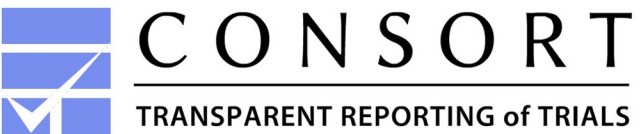

## CONSORT 2010 Flow Diagram

**Enrollment**

Assessed for eligibility (n=152)

Excluded (n=132)
- ◆ Not meeting inclusion criteria (n=96)
- ◆ No longer interested (n=6)
- ◆ Lost to follow up (n=9)
- ◆ Duplicate (n=20)

Enrolled in the study (n=20)

**Commencement**

Commenced and completed control period (4 weeks) (n=20)

Commenced intervention period (n=20)

Discontinued intervention (n=4)
- ◆ pre-existing mental health issues (n=1)
- ◆ difficulties achieving required insulin adjustments (n=1)
- ◆ Lack of time (n=1)
- ◆ Lost to follow up (n=1)

**Follow-Up**

Completed intervention period (12 weeks) (n=16)

**Analysis**

Completers – primary analysis (n=16)

Intention to treat - secondary analysis (n=20)

**Fig 1. CONSORT flow diagram.**

**Table 1. Baseline characteristics of included participants.**

| | Completers (n = 16) | Dropouts (n = 4) |
|---|---|---|
| **Age** (years) | 42.8 ± 13.9 | 36.8 ± 11.0 |
| **Female** | 8 (50%) | 3 (75%) |
| *Ethnicity* | | |
| White | 14 (88%) | 4 (100%) |
| Asian | 1 (6%) | 0 (0%) |
| More than one race | 1 (6%) | 0 (0%) |
| **Born in Australia** | 14 (88%) | 2 (50%) |
| **Body mass index** (kg/m$^2$) | 31.8 ± 5.7 | 24.5 ± 3.4 |
| **Years with diabetes** | 21.2 ± 10.4 | 20.3 ± 17.3 |
| ≥10 years with diabetes | 14 (75%) | 3 (75%) |
| ≥20 years with diabetes | 9 (56%) | 2 (50%) |
| ≥30 years with diabetes | 3 (19%) | 1 (25%) |
| **Use of continuous glucose monitor** | 11 (69%) | 3 (75%) |
| Freestyle Libre | 8 (50%) | 3 (75%) |
| Dexcom G6 | 2 (13%) | 0 (0%) |
| Medtronic Minimed 640G | 1 (6%) | 0 (0%) |
| **Use of insulin pump** | 8 (50%) | 2 (50%) |
| **Multiple daily injections** | 8 (50%) | 2 (50%) |
| **Other medical condition(s)** | 9 (56%) | 2 (50%) |
| Depression/anxiety | 2 (13%) | 1 (25%) |
| Hypertension | 2 (13%) | 1 (25%) |
| Coeliac disease | 2 (13%) | 0 (0%) |
| Irritable bowel syndrome | 1 (6%) | 0 (0%) |
| Inflammatory bowel disease | 0 (0%) | 1 (25%) |
| Rheumatoid arthritis | 1 (6%) | 0 (0%) |
| Other[a] | 3 (19%) | 0 (0%) |
| **Use of other medication[b]** | 11 (69%) | 2 (50%) |

Data presented as means ± standard deviations (rounded to 1 decimal place) or frequency (percent of total) (rounded to nearest whole number).

[a]Other medical condition not listed in exclusion/criteria.

[b]Medication other than insulin.

(P<0.01) and 21 ± 8% (P<0.001) from post-control to post-intervention, with no significant differences between pre- and post-control timepoints. There were no statistically significant differences in diet satisfaction scores between timepoints (P = 0.497) (Table 3), but 10 participants reported 100% diet satisfaction at the post-intervention timepoint compared to only four and five participants at the pre-control and post-control timepoints, respectively.

## HbA1c

HbA1c (%) levels differed between timepoints (P<0.001), with a statistically significant reduction of 0.6 ± 0.7% from post-control to post-intervention (P = 0.003) and no significant difference between pre- and post-control (P = 0.754) (Table 3). Similarly, ITT analysis (n = 20) showed a statistically significant reduction in HbA1c during the intervention period, and no change during the control period (S7 Table). Individual participant changes in HbA1c are shown in Fig 2 (n = 16). Ten participants (63%) experienced a reduction in HbA1c of ≥0.5%

**Table 2. Dietary intake of participants during control and intervention periods.**

| | Pre-control (week -4)[a] | Post-control (week 0) | Post-intervention (week 12) |
|---|---|---|---|
| **Total energy** (kJ/day) | 9817.2 (2620.6) | 9506.0 (2408.7) | 8114.4 (2632.3)[†] |
| **Total energy** (Cal/day) | 2345.3 (626.1) | 2270.9 (575.3) | 1938.6 (628.9)[†] |
| **Carbohydrates** (g/day) | 217.7 (74.5) | 213.6 (67.5) | 63.1 (51.4)[‡] |
| **Carbohydrates** (%TEI) | 34.3 (7.1) | 35.9 (8.3) | 12.5 (8.1)[‡] |
| **Proteins** (g/day) | 103.1 (27.9) | 103.5 (32.5) | 118.1 (27.6) |
| **Proteins** (%TEI) | 18.3 (4.2) | 18.8 (4.9) | 25.6 (4.8)[‡] |
| **Fats** (g/day) | 95.9 (26.4) | 95.9 (33.7) | 130.2 (56.5)[†] |
| **Fats** (%TEI) | 36.4 (5.7) | 36.9 (5.7) | 57.9 (9.2)[‡] |
| **Saturated fats** (g/day) | 35.9 (14.3) | 35.5 (13.7) | 66.1 (55.8)[*] |
| **Saturated fats** (%TEI) | 13.9 (4.6) | 13.6 (2.9) | 24.6 (6.5)[‡] |
| **Fibre** (g/day) | 24.4 (7.0) | 22.5 (7.6) | 18.8 (8.0) |
| **Alcohol** (g/day)^ | 15.0 (38.0) | 8.0 (27.0) | 0.0 (6.0) |

Data presented for n = 16 (completers). Data for pre-control, post-control, and post-intervention timepoints presented as means and standard deviations or medians and interquartile ranges (indicated by ^). Abbreviations–kJ, kilojoules, Cal, Calories; g, grams; TEI, total energy intake.

[*]P<0.025,

[†]P<0.01, and

[‡]P<0.001; indicates significantly different from post-control.

[a]There were no statistically significant differences between pre- and post-control timepoints for any outcome.

during the intervention period, four participants (25%) experienced a reduction between 0.1–0.4%, and two participants (13%) experienced an increase (Fig 2). Seven participants (44%) had an HbA1c level within the diabetes management target of ≤7.0% (≤53 mmol/mol) at post-intervention, compared to two (13%) and three (19%) at pre- and post-control timepoints, respectively (Fig 2).

## Total daily insulin

TDI significantly reduced by 16 ± 11 U/day from post-control to post-intervention (P<0.001), with no change during the control period (P = 0.520) (Table 3).

## Frequency of hypoglycaemia

Frequency of hypoglycaemia episodes did not differ significantly across the three timepoints (P = 0.569, Table 3).

## Glycaemic variability

TIR (blood glucose: 3.5–10 mmol/L) significantly increased by 16 ± 12% from post-control to post-intervention (P<0.001), with no significant change during the control period (P = 0.324) (Table 3). MAGE and mean blood glucose values differed between timepoints such that post-intervention values were significantly lower than post-control (P<0.001), with no differences between pre- and post-control values (P = 0.110 and P = 0.324, respectively) (Table 3). SDBG values reduced during the control period (-0.4 ± 0.6, P = 0.014) and intervention period (-0.8 ± 0.4, P<0.001) (Table 3).

**Table 3. Main clinical outcomes of participants during control and intervention periods.**

|  | Pre-control (week -4) | Post-control (week 0) | Post-intervention (week 12) |
|---|---|---|---|
| *Glycaemic Control*[a] |  |  |  |
| **HbA1c** (%) | 7.7 (0.5) | 7.7 (0.5) | 7.1 (0.7)[†] |
| **HbA1c** (mmol/mol) | 60.7 (5.0) | 60.8 (5.8) | 54.4 (7.5)[†] |
| **Fasting blood glucose** (mmol/L) | 8.6 (3.2) | 8.8 (2.9) | 6.1 (2.1)[†] |
| **Time in range** (%) | 55.1 (14.7) | 58.6 (15.6) | 74.3 (18.1)[‡] |
| **Mean glucose** (mmol/L) | 9.7 (1.4) | 9.3 (1.5) | 8.0 (1.7)[‡] |
| **MAGE** (mmol/L) | 8.1 (1.6) | 7.3 (0.9) | 5.3 (1.5)[‡] |
| **Standard deviation of blood glucose** | 3.2 (0.6)[*] | 2.8 (0.4) | 2.0 (0.5)[‡] |
| **Hypo frequency** (episodes/day)^ | 0.3 (0.3) | 0.3 (0.3) | 0.4 (0.7) |
| **Total daily insulin** (units/day) | 66.3 (22.3) | 65.2 (23.2) | 49.0 (20.8)[‡] |
| *Anthropometry* |  |  |  |
| **Body mass index** (kg/m$^2$) | 31.8 (5.7) | 31.9 (5.9) | 31.1 (5.6)[*] |
| **Body weight** (kg) | 93.8 (18.2) | 93.8 (18.7) | 91.4 (17.7)[*] |
| **Waist circumference** (cm) | 102.8 (17.3) | 103.3 (15.8) | 100.9 (15.3) |
| **Systolic BP** (mmHg) | 123.2 (14.7) | 126.7 (12.1) | 126.7 (17.3) |
| **Diastolic BP** (mmHg) | 76.0 (6.6) | 71.8 (6.9) | 74.1 (6.3) |
| *Lipids* |  |  |  |
| **Total cholesterol** (mmol/L) | 4.6 (0.9) | 4.5 (0.7) | 4.8 (1.2) |
| **HDL cholesterol** (mmol/L) | 1.5 (0.5) | 1.5 (0.5) | 1.6 (0.4) |
| **LDL cholesterol** (mmol/L)^ | 2.5 (0.9) | 2.4 (0.6) | 2.5 (1.8) |
| **Triglycerides** (mmol/L) | 1.1 (0.5) | 1.1 (0.4) | 0.8 (0.2) |
| **Diet satisfaction** (% satisfied)^ | 78.6 (64.3) | 71.4 (50.0) | 100.0 (50.0) |
| **Diabetes quality of life**[b] | 35.0 (7.3) | 33.8 (5.8) | 30.3 (7.4)[*] |

Data presented for n = 16 (completers), except body mass index and waist circumference (n = 13) and systolic BP and diastolic BP (n = 12) due to missing data at some timepoints. Data presented as means and standard deviations or medians and interquartile ranges (indicated by ^). Abbreviations: HbA1c, glycated haemoglobin; MAGE, mean amplitude of glycaemic excursions; hypo, hypoglycaemia; BP, blood pressure; HDL, high density lipoprotein; LDL, low density lipoprotein.

[*]P<0.025,

[†]P<0.01, and

[‡]P<0.001; indicates significantly different from post-control.

[a]If the level of significance was adjusted for multiple endpoints directly measuring glycaemic control (P<0.0042), it is confirmed that the changes in all outcomes (HbA1c, fasting blood glucose, time in range, mean glucose, MAGE, and standard deviation of blood glucose) between the post-control and post-intervention timepoints would remain statistically significant.

[b]A lower score implies a more satisfactory quality of life.

## Diabetes-related quality of life

DQoL scores differed between timepoints (P<0.001), with lower scores at post-intervention compared to post-control (P = 0.015), and no significant difference during the control period (P = 0.289) (Table 3).

## Medication changes

Of the participants taking medications other than insulin at baseline (S10 Table), during the intervention period, one participant reduced their anti-hypertensive medication dosage by 50% and one participant ceased taking anti-hypertensive medications. Of note, all participants reported to receive at least one dose of the COVID-19 vaccination either before or during participation in the study.

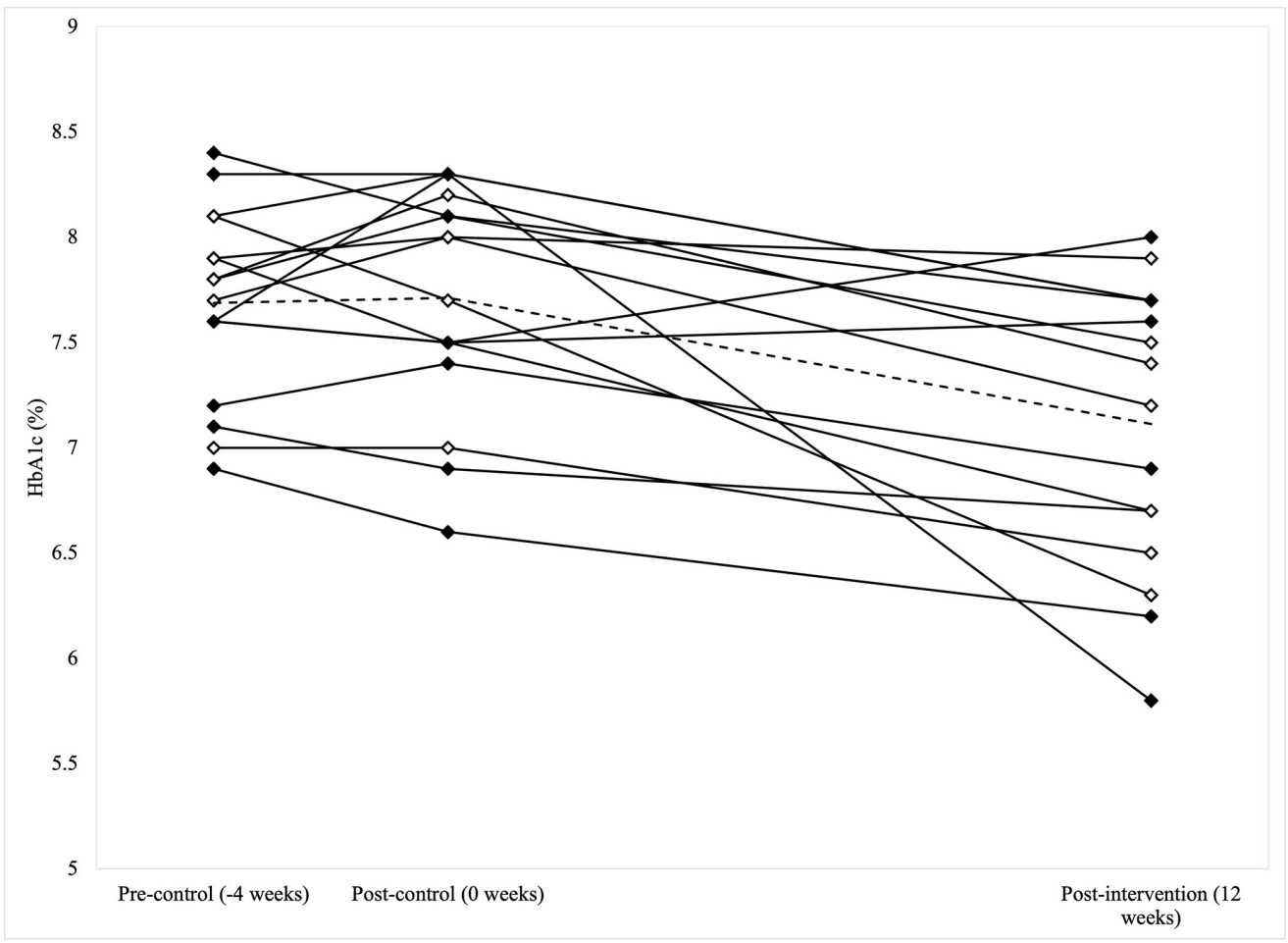

**Fig 2. Glycated haemoglobin (%HbA1c) levels of participants with type 1 diabetes during control and intervention periods.** Data presented for n = 16 (completers). ♦ male; ◇ female; - - - - group mean.

## Additional outcomes

Body weight and BMI reduced between the post-control and post-intervention timepoints (P<0.025), but not between pre- and post-control (Table 3). There was no difference between timepoints for waist circumference, blood pressure, lipid profile, kidney function or liver function (Table 3, S11 Table). Creatine kinase increased by 32 ± 119 U/L between the post-control and post-intervention timepoints (P = 0.008), with no significant change during the control period (P = 0.222) (S11 Table). There were no statistically significant differences in total PAL (P = 0.829), or time spent sitting (P = 0.651) between timepoints (S9 Table).

## Safety monitoring & adverse events

Of the completers, two participants reported ketone levels >0.6 mmol/L during the intervention period but did not require medical treatment: one participant was unwell due to suspected food poisoning, and the other participant had ketones ~3.0 mmol/L in the context of normal blood glucose levels with no negative symptoms. No episodes of severe hypoglycaemia requiring hospitalisation or other adverse events were reported during the intervention period. One

participant who dropped out of the study was hospitalised for ketoacidosis during the control period prior to making any diet or insulin changes.

## Discussion

This single-arm longitudinal intervention study showed that a healthcare professional supported 12-week LC diet (25–75 g/day) improved markers of blood glucose control and quality of life with reduced total daily insulin dosages and no reported episodes of ketoacidosis or severe hypoglycaemia. These preliminary results suggest that a professionally supported LC diet may be a safe and effective approach for T1D management, yet larger randomised controlled trials are needed to confirm our findings.

The 0.6% reduction in mean HbA1c levels observed during the intervention period is considered relatively large and clinically significant, considering that post-intervention HbA1c levels were 7.1% (54 mmol/mol) with 44% of participants reaching the diabetes target of HbA1c ≤7.0% (≤53 mmol/mol) [3, 39]. The present results are similar to previous studies in T1D that reported HbA1c reductions from 7.9 to 7.2% (63 to 55 mmol/mol) with a 12-week LC diet (~100 g/day of carbohydrates) (n = 5) [21] and from 7.6 to 6.9% (60 to 52 mmol/mol) on a 4-year LC diet (≤75 g/day) (n = 48) [22]. A 1% reduction in HbA1c has been reported to reduce the risk of death related to diabetes by 21%, myocardial infarction by 14%, and microvascular complications by 37% [40]. However, HbA1c reductions as low as 0.3% are considered clinically meaningful for reducing diabetes-related complications over the long-term [41, 42]. In the present study, concurrently with the HbA1c reductions, total daily insulin reduced (65 to 49 U/day). This is comparable to results reported by both Krebs (66 to 44 U/day) and Nielsen (43 to 32 U/day) [21, 22]. Data from the Diabetes Control and Complications Trial showed an increased risk of severe hypoglycaemia and excessive weight gain with intensive insulin therapy highlighting the necessity for interventions that achieve glycaemic control without reliance on large insulin doses [43–45].

GV is increasingly recognised as a useful indicator for risk of micro- and macrovascular complications in T1D [46–48]. The current study showed improvements in GV metrics after the LC diet intervention. In the only other known study to have examined the effects of a LC diet on GV in individuals with T1D for 12 weeks, Schmidt et al. [24] showed SDBG was lowered with a LC diet (<100 g/day) compared to a HC diet (>250 g/day); however no differences in TIR (65 and 69% respectively within 3.9–10 mmol/L) were observed. In the present study, the mean TIR (3.5–10 mmol/L) of 74% after the LC intervention was significantly higher compared when to when a higher-carbohydrate diet was consumed during the control period (TIR: 55–59%). The specific reason(s) for the discrepant findings is not clear; however, participants in the present study had lower TIR values at baseline and followed their habitual diet during the control period, whereas participants in the previous study followed a supported HC diet intervention which may have yielded better glycaemic control [24]. The level of carbohydrate intake in the present study was also lower (25–75 g/day vs. 100 g/day), suggesting the possibility that the degree of improvement in TIR could be related to the degree of carbohydrate restriction. Dose-response effects have been reported in T2D patients, showing lower HbA1c levels with greater carbohydrate restriction [18]. Future studies should be conducted to confirm the present findings and to identify the specific factor/s responsible for improved TIR in T1D participants following a LC diet.

Interestingly, a modest but clinically significant reduction in body weight (3%, -2.4 kg) during the 12-week LC diet was observed. This is consistent with LC diet studies conducted in T2D populations [17, 49]. Sustained weight reductions of 2–5% have shown significant benefits in improving CVD risk factors, including SBP and serum triglycerides [50]. The exact

reason(s) for the observed weight reduction cannot be determined from the available data. Previous research report increased satiety levels with adherence to LC diets [51–54] that may lead to spontaneous reductions in total energy intake despite *ad libitum* energy prescriptions. This is supported by the dietary intake data that showed a 15% reduction in reported energy intake during the LC diet intervention (-1392 kJ/day). Another body of evidence demonstrates that insulin can suppress lipolysis [55, 56], and reductions in circulating insulin levels may influence long-term adiposity and body weight change [43, 57, 58]. Interventions that promote adiposity reduction are important in the T1D population, with obesity rates rising rapidly over recent years [59]. It is possible that LC diets could provide an effective weight management strategy for individuals living with T1D who are also overweight or obese. Future studies should examine the long-term effects (>12 weeks) of LC diets on weight management and diet sustainability in T1D. Further, randomised controlled trials which aim to control for the potential effect(s) of weight loss on glycaemic control outcomes under energy balance conditions are also worthy of consideration.

There are several often cited potential safety concerns regarding the use of LC diets including increased risk of hypoglycaemia in T1D [26, 60, 61] and negative impact(s) on quality of life [62]. Conversely, the current study showed no difference in the number of hypoglycaemic episodes between the LC diet intervention and the control phase when a habitual HC diet was consumed, and quality of life improved more during the LC diet. Previous research in T1D also showed that lower-carbohydrate diets reduced episodes of severe hypoglycaemia [63], and a recent randomised crossover trial demonstrated less time spent with blood glucose <3.9 mmol/L during adherence to a LC diet compared to a HC diet [24]. Similar to the present findings, a pilot study of adults living with T1D also reported that a LC diet (<100 g/day) led to improvements in glycaemic control without negatively impacting quality of life [64]. However, previous studies examining the effect(s) of LC diets on T1D management did not measure changes in quality of life [20]. Collectively, the data suggests that LC diet may reduce hypoglycaemia and improve quality of life in individuals with T1D; however, further well-controlled studies are required to confirm these effects.

Improvements in glycaemic control observed with LC diets are often dismissed by concomitant increases in LDL-cholesterol (LDL) levels reported in some [65–69], but not all studies [70, 71]. In the present study, no significant differences in LDL were observed between the control and LC diet phases, despite substantial increases in self-reported saturated fat intake, which has been shown to elevated LDL levels [72–74]. It is worth noting that many participants were taking cholesterol-lowering medication which may have minimised potential increases in LDL. Nevertheless, the strength of LDL as a predictor of CVD remains heavily debated [75], and a recent systematic review showed no association between total LDL and mortality in older adults [76]. Measuring lipid subfractions, rather than total LDL, is an important consideration for future research as the amount of small-dense LDL may be a more reliable marker of CVD risk [77, 78]. At this time, it remains prudent that healthcare practitioners consider the varying impact of LC diets on cholesterol levels, and closely monitor CVD risk using a range of health markers including HbA1c, GV and body weight.

A strength of the current study is that it was conducted in a real-world remote care setting that improves translatability into clinical practice. The intervention was delivered by a registered dietitian via remote contact, and participants were not required to travel outside of their local area for outcome measurements. With consideration of the recent transition toward Telehealth services [79], and the potential for enhanced accessibility to healthcare for people living outside of metropolitan areas and/or those with limited time to attend health clinics, evidence-based remote care interventions have important relevance in the current health climate.

Additionally, the relatively broad eligibility criteria that facilitated inclusion of individuals with common comorbidities (obesity, dyslipidaemia, hypertension, and depression) increases the translatability of these findings, although it should be acknowledged that this may have introduced greater heterogeneity of the data collected. It has also been reported that the COVID-19 pandemic was associated with increased body weight and poorer mental health, likely resulting from changes to work environments, lifestyle behaviours, and eating habits [80, 81]. The present findings suggest the LC diet intervention implemented in this study was able to counteract these effect(s).

The study has several limitations. The single-arm longitudinal study design precludes the ability to directly compare the effects of a LC diet intervention to a HC diet intervention, and the relatively small sample size and short follow-up period (12 weeks) reduces generalisability of the present findings. However, while not a randomised controlled trial, the current within-participant controlled intervention study assists to minimise the potential influences of inherent individual differences of diet and non-diet factors that influence T1D management and blood glucose levels [82–87]. Future studies should use a randomised crossover design with larger participant numbers and a longer follow-up period (>12 weeks) to assess differences between LC and HC diet interventions for T1D management, improve generalisability of results, and to determine whether intervention effects are maintained. Further, lack of resources restricted health-professional support provisions to dietitian-only with limited support from a diabetes educator to provide specific insulin titration advice. Engagement from the participants' usual healthcare practitioners was low and participants experienced challenges managing required changes to basal and bolus insulin dosages whilst following the LC diet. Additionally, some participants experienced difficulties managing conflicting dietary advice from other healthcare professionals. An online survey of 316 individuals with T1D reported that the majority of participants did not feel supported by their healthcare team to follow a very LC diet [88]. A previous systematic review of adults with T2D showed that effective LC diet studies included moderate to high frequency of follow-up with medical professionals [31], that may also be an integral feature of other successful dietary interventions [89, 90]. For patients to achieve improved health outcomes with a multidisciplinary team approach, upskilling of healthcare practitioners may be required to increase confidence in managing patients using a range of safe and effective dietary approaches, which could include LC diets. With that said, a high frequency of follow-up with healthcare professionals (i.e., once every fortnight), as used in the current study, may not be widely accessible and future researchers may wish to develop and evaluate strategies aimed at supporting patients between consultations to promote safety and adherence.

In the present study, the LC diet intervention prescribed the consumption of minimally-processed whole foods and to minimise the intake of ultra-processed foods. However, due to the preliminary nature of this single arm study that did not assess diet quality, it is difficult to determine the specific effect(s) of changes in diet quality on the favourable metabolic changes observed. Prioritisation of minimally-processed foods has been identified as a core feature of safe and effective LC diet interventions used in T2D management [31]. In addition, common health-promoting dietary patterns, such as the Mediterranean diet, also prioritise consumption of minimally-processed foods such as dairy, nuts, seeds, legumes, meat, fish, and eggs, while limiting ultra-processed foods [91, 92]. Future randomised controlled studies should aim to control for the potential impact(s) of diet quality on T1D management outcomes when comparing LC diets with diets higher in carbohydrates. Nevertheless, it would be prudent to consider diet quality in the design and delivery of LC diets in clinical practice.

## Conclusions

These preliminary findings suggest that a professionally supported LC diet may improve markers of blood glucose control and quality of life with reduced exogenous insulin requirements and no evidence of increased risk of hypoglycaemia or ketoacidosis in adults with T1D. Given the potential benefits of this intervention, larger, longer-term randomised controlled trials are needed to confirm these findings and examine clinical endpoints to better demonstrate the efficacy of LC diets in T1D management.

## Supporting information

**S1 File. Study protocol (version 3).**
(PDF)

**S1 Table. TREND checklist.**
(PDF)

**S2 Table. Sample meal plan A for low-carbohydrate diet (carbs: 25 g/day).** Abbreviations: carbs, total dietary carbohydrates; g, grams; tsp, teaspoon; tbs, tablespoon. *Cooked weight; ^natural nut butter (nuts and salt only). Other instructions: If you want to add snacks to this meal plan and your carbohydrate target is 25 g/day, then your snacks should be proteins and/ or fats that do not also contain carbs. If you want to increase your portions of proteins and/or fats at meals to reach satiety, you can.
(DOCX)

**S3 Table. Sample meal plan B for low-carbohydrate diet (carbs: 50 g/day).** Abbreviations: carbs, total dietary carbohydrates; g, grams; tsp, teaspoon; tbs, tablespoon. *Cooked weight; ^natural nut butter (nuts and salt only). Other instructions: If you want to add snacks to this meal plan and your carbohydrate target is 50 g/day, then your snacks should be proteins and/ or fats that do not also contain carbs. If you want to increase your portions of proteins and/or fats at meals to reach satiety, you can.
(DOCX)

**S4 Table. Sample meal plan C for low-carbohydrate diet (carbs: 60 g/day).** Abbreviations: carbs, total dietary carbohydrates; g, grams; tsp, teaspoon; tbs, tablespoon. *Cooked weight; ^natural nut butter (nuts and salt only). Other instructions: If your carbohydrate target is 60–75 g/day of total carbohydrates, you may add 1–2 snacks containing 5–10 g of carbohydrates to this meal plan. If you want to increase your portions of proteins and/or fats at meals to reach satiety, you can.
(DOCX)

**S5 Table. Glycaemic variability data with exclusions (n = 15).** Data presented for n = 15 with exclusions applied to the raw data (days that had <80% complete values were excluded) and participants with <3 days of complete data were excluded (n = 1). Data presented as means and standard deviations or medians and interquartile ranges (indicated by ^). *P<0.025, †P<0.01, and ‡P<0.001; indicates significantly different from post-control.
(DOCX)

**S6 Table. Dietary intake for participants with type 1 diabetes during control and intervention periods (intention-to-treat).** Data presented for n = 20 (intention to treat). Data presented as means and standard deviations or medians and interquartile ranges (indicated by ^). * = P<0.025 between timepoints (post-control and pre-control or post-intervention and post-control). Abbreviations–int, intervention; kJ, kilojoules, Cal, Calories; g, grams; TEI, total

energy intake.
(DOCX)

**S7 Table. Main clinical outcomes for participants with type 1 diabetes during control and intervention periods (intention-to-treat).** Data presented for n = 20 (intention to treat), except systolic BP and diastolic BP (n = 19) due to missing data at >1 timepoints. Data presented as means and standard deviations or medians and interquartile ranges (indicated by ^). * = P<0.025 between timepoints (post-control and pre-control or post-int and post-control). #A lower score implies a more satisfactory quality of life. Abbreviations–HbA1c, glycated haemoglobin; MAGE, mean amplitude of glycaemic excursions; hypo, hypoglycaemia; BP, blood pressure; HDL, high density lipoprotein; LDL, low density lipoprotein.
(DOCX)

**S8 Table. Additional outcomes for participants with type 1 diabetes during control and intervention periods (intention-to-treat).** Data presented for n = 20 (intention to treat), except uric acid (n = 19) due to missing data at >1 timepoints. Data presented as means and standard deviations or medians and interquartile ranges (indicated by ^). * = P<0.025 between timepoints (post-control and pre-control or post-intervention and post-control).
(DOCX)

**S9 Table. Total physical activity and time spent sitting for participants with type 1 diabetes during control and intervention periods.** Data presented as means and standard deviations or medians and interquartile ranges (indicated by ^). * = P<0.025 between timepoints (post-control and pre-control or post-intervention and post-control). Abbreviations–PAL, physical activity level; MET, metabolic equivalent of task; min, minutes, ITT, intention to treat.
(DOCX)

**S10 Table. Use of medications other than insulin by participants with type 1 diabetes at baseline.** *data taken from screening survey.
(DOCX)

**S11 Table. Additional outcomes for participants with type 1 diabetes during control and intervention periods (n = 16).** Data presented for n = 16 (completers), except calcium (n = 15), corrected calcium (n = 15), phosphate (n = 15), uric acid (n = 15), magnesium (n = 15), creatine kinase (n = 15), and body weight (n = 13) due to missing data at ≥1 timepoint. Data for pre-control, post-control, and post-intervention timepoints presented as means and standard deviations or medians and interquartile ranges (indicated by ^). * = P<0.025. #A lower score implies a more satisfactory quality of life.
(DOCX)

## Acknowledgments

Medtronic Australasia donated the necessary Continuous Blood Glucose Monitoring sensors and devices for participants throughout the study.

Dr Lim and Dr Lee provided in-kind time to perform the roles of the study physician and study endocrinologist, respectively.

## Author Contributions

**Conceptualization:** Jessica L. Turton, Grant D. Brinkworth, Helen M. Parker, David Lim, Kevin Lee, Amy Rush, Rebecca Johnson, Kieron B. Rooney.

**Formal analysis:** Jessica L. Turton, Grant D. Brinkworth, Kieron B. Rooney.

**Funding acquisition:** Jessica L. Turton, Grant D. Brinkworth, Kieron B. Rooney.

**Investigation:** Jessica L. Turton, David Lim, Kevin Lee, Amy Rush, Rebecca Johnson.

**Methodology:** Jessica L. Turton, Grant D. Brinkworth, Helen M. Parker, Kieron B. Rooney.

**Project administration:** Jessica L. Turton.

**Resources:** Jessica L. Turton, David Lim, Kevin Lee, Amy Rush, Rebecca Johnson.

**Supervision:** Grant D. Brinkworth, Helen M. Parker, Kieron B. Rooney.

**Writing – original draft:** Jessica L. Turton.

**Writing – review & editing:** Jessica L. Turton, Grant D. Brinkworth, Helen M. Parker, David Lim, Kevin Lee, Amy Rush, Rebecca Johnson, Kieron B. Rooney.

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
