## [Decision Letter · Decision Letter 0]

26 Jan 2023

PONE-D-22-32366

Effects of a low-carbohydrate diet in adults with type 1 diabetes management: a single arm non-randomised clinical trial

PLOS ONE

Dear Dr. Turton,

Thank you for submitting your manuscript to PLOS ONE. After careful consideration, we feel that it has merit but does not fully meet PLOS ONE’s publication criteria as it currently stands. Therefore, we invite you to submit a revised version of the manuscript that addresses the points raised during the review process.

We look forward to receiving your revised manuscript.

Kind regards,

Caroline Bull

Staff Editor

PLOS ONE

Journal Requirements:

   "I have read the journal's policy and the authors of this manuscript have the following competing interests: All healthcare practitioners involved in the study (Jessica Turton, Kevin Lee, David Lim, and Amy Rush) have their own private consulting businesses in Australia. The outcome(s) of the intervention may be considered a reflection of their proficiency as healthcare practitioners. Nevertheless, existing patients of Ms Turton, Dr Lee, Dr Lim, and Mrs Rush were excluded from participating in the study. " 

3. We note that the original protocol file you uploaded contains a confidentiality notice indicating that the protocol may not be shared publicly or be published. Please note, however, that the PLOS Editorial Policy requires that the original protocol be published alongside your manuscript in the event of acceptance. Please note that should your paper be accepted, all content including the protocol will be published under the Creative Commons Attribution (CC BY) 4.0 license, which means that it will be freely available online, and any third party is permitted to access, download, copy, distribute, and use these materials in any way, even commercially, with proper attribution.

Therefore, we ask that you please seek permission from the study sponsor or body imposing the restriction on sharing this document to publish this protocol under CC BY 4.0 if your work is accepted. We kindly ask that you upload a formal statement signed by an institutional representative clarifying whether you will be able to comply with this policy. Additionally, please upload a clean copy of the protocol with the confidentiality notice (and any copyrighted institutional logos or signatures) removed.

4. We note that the original protocol that you have uploaded as a Supporting Information file contains an institutional logo. As this logo is likely copyrighted, we ask that you please remove it from this file and upload an updated version upon resubmission.

Additional Editor Comments:

The manuscript has been evaluated by two reviewers, and their comments are available below. 

The reviewers have requested more information on the statistical aspects of the study, specifically regarding the sample size calculations and the adjustment of P values due to the small sample size of the study.

Furthermore, Reviewer 1 in their thorough evaluation of the manuscript has mentioned that the study may not fall within the scope of a Clinical Trial due to the absence of a control group. We would like to note that PLOS ONE follows the WHO definition of a Clinical Trial which states that “a clinical trial is any research study that prospectively assigns human participants or groups of humans to one or more health-related interventions to evaluate the effects on health outcomes. Interventions include but are not restricted to drugs, cells and other biological products, surgical procedures, radiological procedures, devices, behavioural treatments, process-of-care changes, preventive care, etc.”. Please note that this definition includes Phase I to Phase IV trials, and Phase I study routinely include small group of healthy people. In addition to the definition set by the WHO, the ICMJE further clarifies that ‘health-related interventions include any intervention used to modify a biomedical or health-related outcome (for example, drugs, surgical procedures, devices, behavioural treatments, dietary interventions, and process-of-care changes). Health outcomes include any biomedical or health-related measures obtained in patients or participants, including pharmacokinetic measures and adverse events’.

Here we consider that the development of low-carbohydrate diet on glycaemic control has direct clinical implications, and therefore consider that your study fits the above definition of a clinical trial under the WHO definition

Could you please carefully revise the manuscript to address all comments raised? 

Reviewers' comments:

Reviewer's Responses to Questions

**Comments to the Author**

1. Is the manuscript technically sound, and do the data support the conclusions?

Reviewer #1: Partly

Reviewer #2: Yes

2. Has the statistical analysis been performed appropriately and rigorously? 

Reviewer #1: Yes

Reviewer #2: No

3. Have the authors made all data underlying the findings in their manuscript fully available?

Reviewer #1: Yes

Reviewer #2: Yes

4. Is the manuscript presented in an intelligible fashion and written in standard English?

Reviewer #1: Yes

Reviewer #2: Yes

5. Review Comments to the Author

Reviewer #1: Plos ONE 22 32366 reviewer comments.

This paper describes a small, single-centre longitudinal cohort study of 20 adults with suboptimally controlled type 1 diabetes who underwent a 12-week low carbohydrate diet (after a four week standard diet “run-in”, delivered under the guidance of a dietitian via telehealth), 16 of whom completed the intervention and had follow-up measures. Improvements in several important and relevant outcomes were noted at 12 weeks.

This is an excellently written and comprehensive paper, which is very well presented and addresses an interesting and important area of clinical practice. The findings are important. However, they must be seen as entirely preliminary, because the sample size in the study is small, it involves data from just one centre, and (most importantly) there is no control group. It is clear from the quality of the writing of the paper and the supplementary material that the intervention delivery and the characterisation of study participants has been first class. However, having given it careful consideration, I think it is difficult to argue that this really is a clinical trial, because there is no control group with which to make meaningful comparisons. So, technically, this is a longitudinal cohort study. There is nothing wrong with that – it offers an important opportunity to gain insights into effect sizes of the intervention on different outcomes, in order that a subsequent trial (an RCT, with a control group) can be adequately powered. In my opinion, it diminishes the credibility of an excellent study with important preliminary findings to call it a trial and make conclusions that are not justified. I think the conclusions need to be much more measured and conservative, such as stating that the apparent improvements in important clinical outcomes warrant consideration in an RCT. To be clear, it would be wrong to seek to change clinical practice guidelines internationally on the basis of a single study with 12 weeks of follow-up with no control group. What if the improvements are due to the significant reduction in alcohol consumption, for example?

Aside from this fundamental issue, the work is very strong. Some of the referencing at the start of the introduction could be more succinct. I may have missed it but did the authors consider the fact that participants were still consuming 94.6g per day on the low carb group? It would be good to report HbA1c in mmol/mol. Thanks for this important contribution to the field.

Reviewer #2: A follow-up measure may be conducted to evaluate whether the intervention effects will maintain.

What test was used for sample size calculation?

For a small sample size, the normality test will not have sufficient power and therefore more likely to claim the data follows a normal distribution.

The sample size is relatively small for so many secondary endpoints. The p values should be adjusted for multiple secondary endpoints also.

Table 3 Changes in BMI and weight do not seem to be significant. Effect sizes less than 0.3.

Table 3 if no variable was reported with median, then do not mention median in the footnote.

6. PLOS authors have the option to publish the peer review history of their article (what does this mean?). If published, this will include your full peer review and any attached files.

Reviewer #1: **Yes: **Francis Finucane

Reviewer #2: No

---

## [Author Response · Author response to Decision Letter 0]

13 Mar 2023

Editor

Comment 1: Please ensure that your manuscript meets PLOS ONE's style requirements, including those for file naming.

Response: The final manuscript has been reviewed and updated where necessary to ensure the manuscript formatting style is consistent with the style requirements of PLOS ONE.

Comment 2: Thank you for stating the following in the Competing Interests section: "I have read the journal's policy and the authors of this manuscript have the following competing interests: All healthcare practitioners involved in the study (Jessica Turton, Kevin Lee, David Lim, and Amy Rush) have their own private consulting businesses in Australia. The outcome(s) of the intervention may be considered a reflection of their proficiency as healthcare practitioners. Nevertheless, existing patients of Ms Turton, Dr Lee, Dr Lim, and Mrs Rush were excluded from participating in the study. " 

Please confirm that this does not alter your adherence to all PLOS ONE policies on sharing data and materials, by including the following statement: "This does not alter our adherence to PLOS ONE policies on sharing data and materials.” If there are restrictions on sharing of data and/or materials, please state these. Please note that we cannot proceed with consideration of your article until this information has been declared. 

Response: It is confirmed that that the authors’ declared competing interests do not alter adherence to PLOS ONE policies on sharing data and materials and that the additional recommended statement can be added. The updated Competing Interests statement has been included in the cover letter uploaded with this submission.

Comment 3: We note that the original protocol file you uploaded contains a confidentiality notice indicating that the protocol may not be shared publicly or be published. Please note, however, that the PLOS Editorial Policy requires that the original protocol be published alongside your manuscript in the event of acceptance. Please note that should your paper be accepted, all content including the protocol will be published under the Creative Commons Attribution (CC BY) 4.0 license, which means that it will be freely available online, and any third party is permitted to access, download, copy, distribute, and use these materials in any way, even commercially, with proper attribution.

Therefore, we ask that you please seek permission from the study sponsor or body imposing the restriction on sharing this document to publish this protocol under CC BY 4.0 if your work is accepted. We kindly ask that you upload a formal statement signed by an institutional representative clarifying whether you will be able to comply with this policy. Additionally, please upload a clean copy of the protocol with the confidentiality notice (and any copyrighted institutional logos or signatures) removed.

Response: It is confirmed that we have permission from the study sponsor (The University of Sydney) to share and publish our study protocol, and to do so with all institutional logos, signatures, and confidentiality notices removed. A formal statement signed by an institutional representative has been uploaded with this submission. 

Comment 4: We note that the original protocol that you have uploaded as a Supporting Information file contains an institutional logo. As this logo is likely copyrighted, we ask that you please remove it from this file and upload an updated version upon resubmission.

Response: It is confirmed that institutional logos have been removed from the study protocol and the revised document has been uploaded with this submission.

Referee 1

Comment 1: This paper describes a small, single-centre longitudinal cohort study of 20 adults with suboptimally controlled type 1 diabetes who underwent a 12-week low carbohydrate diet (after a four week standard diet “run-in”, delivered under the guidance of a dietitian via telehealth), 16 of whom completed the intervention and had follow-up measures. Improvements in several important and relevant outcomes were noted at 12 weeks.

This is an excellently written and comprehensive paper, which is very well presented and addresses an interesting and important area of clinical practice. The findings are important. However, they must be seen as entirely preliminary, because the sample size in the study is small, it involves data from just one centre, and (most importantly) there is no control group. It is clear from the quality of the writing of the paper and the supplementary material that the intervention delivery and the characterisation of study participants has been first class. However, having given it careful consideration, I think it is difficult to argue that this really is a clinical trial, because there is no control group with which to make meaningful comparisons. So, technically, this is a longitudinal cohort study. There is nothing wrong with that – it offers an important opportunity to gain insights into effect sizes of the intervention on different outcomes, in order that a subsequent trial (an RCT, with a control group) can be adequately powered. In my opinion, it diminishes the credibility of an excellent study with important preliminary findings to call it a trial and make conclusions that are not justified. 

I think the conclusions need to be much more measured and conservative, such as stating that the apparent improvements in important clinical outcomes warrant consideration in an RCT. To be clear, it would be wrong to seek to change clinical practice guidelines internationally on the basis of a single study with 12 weeks of follow-up with no control group. What if the improvements are due to the significant reduction in alcohol consumption, for example?

Response: The authors thank the reviewer for the positive review. The authors agree that the lack of a control group and small sample size prevents definitive conclusions about the effect(s) of the low-carbohydrate diet on type 1 diabetes outcomes from being made, and that larger randomised controlled trials (RCTs) are needed to confirm the present findings. The final sentence of the conclusion states (page 27, lines 527-529): “Larger, longer-term randomised controlled trials are needed to confirm these findings and to examine clinical endpoints to better understand the applicability of LC diets for the management of T1D.” In addition, changes throughout the manuscript have been made to further acknowledge the limitations of the study design used, the necessity for larger RCTs, and to soften the strength of any conclusions being made:

i) The abstract conclusions have been updated (pages 2-3, lines 37-41): “This study demonstrated that a professionally supported LC diet improved markers of blood glucose control and quality of life, reduced exogenous insulin requirements, with no evidence of increased hypoglycaemia or ketoacidosis risk in adults with T1D. This suggests that LC diets may be a feasible approach for T1D management in adults, yet larger randomised controlled trials are needed to confirm these findings.”

ii) The first paragraph of the discussion has been updated (page 21-22, lines 392-397). “This single-arm longitudinal intervention study showed that a healthcare professional supported 12-week LC diet (25-75 g/day) improved markers of blood glucose control and quality of life with reduced total daily insulin dosages and no reported episodes of ketoacidosis or severe hypoglycaemia. These preliminary results suggest that a professionally supported LC diet may be a feasible and safe approach for T1D management, yet larger randomised controlled trials are needed to confirm our findings.”

iii) The following text in the limitations paragraph of the discussion has been updated (page 26, lines 496-505): “The single-arm longitudinal study design precludes the ability to directly compare the effects of a LC [low-carbohydrate] diet intervention to a HC [high-carbohydrate] diet intervention, and the relatively small sample size and short follow-up period (12 weeks) reduces generalisability of the present findings. However, while not a randomised controlled trial, the current within-participant controlled intervention study assists to minimise the potential influences of inherent individual differences of diet and non-diet factors that influence T1D management and blood glucose levels.[83-88] Future studies should use a randomised crossover design with larger participant numbers and a longer follow-up period (>12 weeks) to assess differences between LC and HC diet interventions for T1D management, improve generalisability of results, and to determine whether intervention effects are maintained.”

In addition, we confirm the present study design falls within the scope and definition of a clinical trial, and this has been confirmed by the Editor at PLOS ONE. According to the World Health Organization (WHO), “A clinical trial is any research study that prospectively assigns human participants or groups of humans to one or more health-related interventions to evaluate the effects on health outcomes. Interventions include but are not restricted to drugs, cells and other biological products, surgical procedures, radiological procedures, devices, behavioural treatments, process-of-care changes, preventive care, etc.” Our clinical trial was prospectively registered on the Australian New Zealand Clinical Trials Registry (ANZCTR): https://www.anzctr.org.au/ACTRN12621000764831.aspx. 

Comment 2: Aside from this fundamental issue, the work is very strong. Some of the referencing at the start of the introduction could be more succinct. I may have missed it but did the authors consider the fact that participants were still consuming 94.6g per day on the low carb group? It would be good to report HbA1c in mmol/mol. Thanks for this important contribution to the field.

Response: It is confirmed that the introduction has now been updated to increase brevity and some references have been removed to make this section more succinct (pages 4-5, lines 47-92). The paragraph discussing the efficacy of low-carbohydrate diets for type 2 diabetes has been shortened and combined with the paragraph discussing the previous evidence investigating lower-carbohydrate diets in adults with type 1 diabetes. The length of the introduction has been reduced from three pages to two pages.

It is confirmed that the reported mean carbohydrate intake of participants during the control and intervention periods was presented in the results section and has been considered in the interpretation of the results. The mean reported intake of carbohydrates at the post-intervention timepoint for completers (n=16) was 63.1 g/day, which was within the targeted dietary prescription range (25-75 g/day of digestible carbohydrates) (Table 2, page 16-17). The mean reported intake of carbohydrates at the post-intervention timepoint for the intention-to-treat (ITT) analysis (n=20) was 94.6 g/day. The ITT analysis was calculated conservatively by carrying the last reported value forward (i.e., post-control timepoint) which reflected a higher-carbohydrate dietary intake given that participants were still in the control period at this timepoint. As such, it is inappropriate to draw conclusions about intervention adherence using the dietary intake data from the ITT analysis. Of the four participants who dropped out of the study, reasons for drop-out were reported in the manuscript (lines 264-267, page 14) and included: “pre-existing mental health issues and experienced difficulties meeting the study requirements (n=1), difficulty achieving insulin adjustment requirements (n=1), a lack of time to meet the study requirements (n=1), and loss to follow up (n=1)” (lines 264-267, page 14). Both sets of results (completers and ITT) were presented for full transparency. 

The authors agree that it would be useful to report HbA1c values in mmol/mol, so the manuscript text and Table 3 (page 18) have been revised to include HbA1c values in mmol/mol in parenthesis beside the %HbA1c values, where appropriate. 

Referee 2

Comment 1: A follow-up measure may be conducted to evaluate whether the intervention effects will maintain. Mention this as a limitation of this report and recommendations for further studies.

Response: The authors agree that longer term follow-up data (>12 weeks) assessing the maintenance of the intervention effect(s) is important and further research with longer follow-ups is required. The necessity for long-term follow up assessments in future studies has been highlighted in the manuscript discussion (page 24, lines 448-449) and conclusions (page 27, lines 527-529).

Consistent with the reviewer’s recommendation, acknowledgment of the relatively short follow-up period (12 weeks) as a study limitation has been included in the discussion (page 26, lines 496-505). This section now reads: “The single-arm longitudinal study design precludes the ability to directly compare the effects of a LC diet intervention to a HC diet intervention, and the relatively small sample size and short follow-up period (12 weeks) reduces generalisability of the present findings. However, while not a randomised controlled trial, the current within-participant controlled intervention study assists to minimise the potential influences of inherent individual differences of diet and non-diet factors that influence T1D management and blood glucose levels.[83-88] Future studies should use a randomised crossover design with larger participant numbers and a longer follow-up period (>12 weeks) to assess differences between LC and HC diet interventions for T1D management, improve generalisability of results, and to determine whether intervention effects are maintained.”

Comment 2: What test was used for sample size calculation?

Response: It is confirmed that a paired t-test of comparisons was used to calculate the required sample size. This information has now been added to the Statistical Analyses paragraph of the methods section (page 12, line 248).

Comment 3: For a small sample size, the normality test will not have sufficient power and therefore more likely to claim the data follows a normal distribution.

Response: It is confirmed that the visual assessment of histograms and PP-plots conducted by two investigators were used to assess data normality rather than normality tests. This approach has been described in the Statistical Analyses paragraph of the methods section (page 12, lines 251-252): “Prior to analysis, data was assessed for normality by two investigators (K.B.R. and J.L.T.) using histograms and PP-plots.”

Nevertheless, we agree with the reviewer that it is difficult to assess normality with a small sample size, and this inherent limitation of the present study has now been acknowledged in the manuscript discussion (page 26, lines 496-505). In a prudent approach, two sets of statistical analyses were performed for each primary outcome based on both normal and non-normal distributions and it is confirmed that both approaches produced a similar pattern of results.

Comment 4: The sample size is relatively small for so many secondary endpoints. The p values should be adjusted for multiple secondary endpoints also.

Response: The manuscript has been updated to ensure the small sample size has been acknowledged as a limitation in the discussion and to emphasise the preliminary nature of the present results that need to be confirmed by larger randomised controlled trials. In accordance with a previously published recommended approach, it is confirmed that a primary outcome was specified (HbA1c) rather than adjusting the p values (Feise, R.J. Do multiple outcome measures require p-value adjustment?. BMC Med Res Methodol 2, 8 (2002). https://doi.org/10.1186/1471-2288-2-8). Nonetheless, it is acknowledged that the study included multiple secondary endpoints directly measuring glycaemic control including HbA1c, fasting blood glucose, time in range, mean glucose, MAGE, and standard deviation of blood glucose. If we were to adjust the p-value for these six outcomes, in addition to the adjustment already made for multiple timepoints, the adjusted level of significance would be P<0.0042. Table 3 reports the changes in these glycaemic control outcomes over three timepoints. We can confirm that if the level of significance was P<0.0042 then the changes in all six glycaemic control outcomes between the post-control and post-intervention timepoints would remain statistically significant. Four of six outcomes have a p value of <0.001 (as indicated by the symbol: ‡). The other two outcomes, HbA1c and fasting blood glucose, have p-values of 0.003 and 0.001, respectively. For full transparency, we have added the following information to the Table 3 legend (page 19): “If the level of significance was adjusted for multiple endpoints directly measuring glycaemic control (P<0.0042), it is confirmed that the changes in all outcomes (HbA1c, fasting blood glucose, time in range, mean glucose, MAGE, and standard deviation of blood glucose) between the post-control and post-intervention timepoints would remain statistically significant.”

Comment 5: Table 3 Changes in BMI and weight do not seem to be significant. Effect sizes less than 0.3.

Response: Based on previous studies, the observed changes in BMI and body weight that occurred during the diet intervention period can be considered clinically significant and has been discussed in the manuscript (page 23, lines 435-438): “a modest but clinically significant reduction in body weight (3%, -2.4 kg) during the 12-week LC diet was observed. This is consistent with LC diet studies conducted in T2D populations.[17,50] Sustained weight reductions of 2-5% have shown significant benefits in improving CVD risk factors, including SBP and serum triglycerides.[51]”

Comment 6: Table 3 if no variable was reported with median, then do not mention median in the footnote.

Response: We can confirm that the diet satisfaction values in the second last row of Table 3 (page 19) were presented as medians and interquartile ranges as indicated by ^. Accordingly, the median foot note has been retained.

---

## [Decision Letter · Decision Letter 1]

28 Apr 2023

PONE-D-22-32366R1Effects of a low-carbohydrate diet in adults with type 1 diabetes management: a single arm non-randomised clinical trialPLOS ONE

Dear Dr. Turton,

Thank you for submitting your manuscript to PLOS ONE. After careful consideration, we feel that it has merit but does not fully meet PLOS ONE’s publication criteria as it currently stands. Therefore, we invite you to submit a revised version of the manuscript that addresses the points raised during the review process.

We look forward to receiving your revised manuscript.

Kind regards,

Jennifer Annette Campbell, PHD, MPH

Academic Editor

PLOS ONE

Reviewers' comments:

Reviewer's Responses to Questions

**Comments to the Author**

1. If the authors have adequately addressed your comments raised in a previous round of review and you feel that this manuscript is now acceptable for publication, you may indicate that here to bypass the “Comments to the Author” section, enter your conflict of interest statement in the “Confidential to Editor” section, and submit your "Accept" recommendation.

Reviewer #1: (No Response)

Reviewer #2: All comments have been addressed

Reviewer #3: All comments have been addressed

2. Is the manuscript technically sound, and do the data support the conclusions?

Reviewer #1: Partly

Reviewer #2: (No Response)

Reviewer #3: Yes

3. Has the statistical analysis been performed appropriately and rigorously? 

Reviewer #1: Yes

Reviewer #2: (No Response)

Reviewer #3: Yes

4. Have the authors made all data underlying the findings in their manuscript fully available?

Reviewer #1: Yes

Reviewer #2: (No Response)

Reviewer #3: Yes

5. Is the manuscript presented in an intelligible fashion and written in standard English?

Reviewer #1: Yes

Reviewer #2: (No Response)

Reviewer #3: Yes

6. Review Comments to the Author

Reviewer #1: Overall the authors have made good progress with amending the paper, though I thought that the responses to the extensive reviewer comments could have been more elaborate and substantive. I think it is wrong to conclude, as the authors have done, that "This study demonstrated that a professionally supported LC diet improved markers of blood glucose control and quality of life with reduced exogenous insulin requirements and no evidence of increased risk of hypoglycaemia or ketoacidosis" - If it helps the authors, I would suggest re-writing this to state that "These preliminary findings suggest that a professionally supported LC diet may lead to improvements in markers of blood glucose control and quality of life with reduced exogenous insulin requirements and no evidence of increased risk of hypoglycaemia or ketoacidosis. However a randomised controlled trial would need to be undertaken to demonstrate the efficacy of this intervention to influence these clinical outcomes. Given the potential benefits of this intervention, such a trial is warranted."

For me, amending the level of inference around clinical practice that the authors have made in the conclusions would be essential before this paper would be suitable for publication.

Reviewer #2: (No Response)

Reviewer #3: This is a well-designed study and well written manuscript on the effects of a LC on glycemic control and other health outcomes in type 1 diabetes. The findings of the study are in line with previous observational studies and interventions that showed improvements in glycemic and other health outcomes following a LC diet in type 1 diabetes in adults. Nevertheless, the intervention is innovative given that it was administered remotely by professionals, which also makes it practical and relevant in the current times with the widespread use of telehealth. The authors have adequately addressed the comments made by the reviewers in the first round of review, especially pertaining to the statistical analyses and acknowledging the study’s limitations regarding the small sample size, the short-term testing of the intervention and outcomes and the quasi-experimental design.

Comments:

Authors need to add to the discussion that longer term studies are needed, not only to test the long-term effects of LC in type 2 diabetes, but also to test the LC diet sustainability and its subsequent effects on improved caloric intake and weight loss, which could be mediating factors for improved health outcomes. This is especially important, given that LC diets are known to induce weight loss in the short term only, and that their effects on weight loses its significance in the longer term, compared to other diets.

Additionally, authors need to stress in the discussion on the importance of diet quality when recommending a LC diet for type 1 diabetes management, like emphasizing whole foods and healthy protein and fat sources rather than processed sources, as described in the study protocol (Ref 30: Turton JL, Brinkworth GD, Parker HM, Lee K, Lim D, Rush A, et al. Effects of a low-carbohydrate diet in adults with type 1 diabetes: an interventional study protocol. 2021.8(3):11. Epub 2021-07-22. doi: 10.18203/2349-3259.ijct20212846.), which might also be another contributing factor for the favorable health outcomes.

7. PLOS authors have the option to publish the peer review history of their article (what does this mean?). If published, this will include your full peer review and any attached files.

Reviewer #1: No

Reviewer #2: No

Reviewer #3: No

---

## [Author Response · Author response to Decision Letter 1]

10 May 2023

Reviewer 1

Comment 1: Overall the authors have made good progress with amending the paper, though I thought that the responses to the extensive reviewer comments could have been more elaborate and substantive. I think it is wrong to conclude, as the authors have done, that "This study demonstrated that a professionally supported LC diet improved markers of blood glucose control and quality of life with reduced exogenous insulin requirements and no evidence of increased risk of hypoglycaemia or ketoacidosis" - If it helps the authors, I would suggest re-writing this to state that "These preliminary findings suggest that a professionally supported LC diet may lead to improvements in markers of blood glucose control and quality of life with reduced exogenous insulin requirements and no evidence of increased risk of hypoglycaemia or ketoacidosis. However a randomised controlled trial would need to be undertaken to demonstrate the efficacy of this intervention to influence these clinical outcomes. Given the potential benefits of this intervention, such a trial is warranted." For me, amending the level of inference around clinical practice that the authors have made in the conclusions would be essential before this paper would be suitable for publication.

Response: The authors thank this reviewer for their thorough review and constructive comments. It is confirmed that the conclusion has now been updated in the Abstract and the main text Conclusions.

The respective section of the Abstract (page 2, lines 34-39) now reads as:

“These preliminary findings suggest that a professionally supported LC diet may lead to improvements in markers of blood glucose control and quality of life with reduced exogenous insulin requirements and no evidence of increased hypoglycaemia or ketoacidosis risk in adults with T1D. Given the potential benefits of this intervention, larger, longer-term randomised controlled trials are warranted to confirm these findings.”

The respective section of the main text Conclusions (page 28, lines 556-561) now reads as:

“These preliminary findings suggest that a professionally supported LC diet may improve markers of blood glucose control and quality of life with reduced exogenous insulin requirements and no evidence of increased risk of hypoglycaemia or ketoacidosis in adults with T1D. Given the potential benefits of this intervention, larger, longer-term randomised controlled trials are needed to confirm these findings and examine clinical endpoints to better demonstrate the efficacy of LC diets in T1D management.”

Reviewer 2

(No comments to address)

Reviewer 3

Comment 1: This is a well-designed study and well written manuscript on the effects of a LC on glycemic control and other health outcomes in type 1 diabetes. The findings of the study are in line with previous observational studies and interventions that showed improvements in glycemic and other health outcomes following a LC diet in type 1 diabetes in adults. Nevertheless, the intervention is innovative given that it was administered remotely by professionals, which also makes it practical and relevant in the current times with the widespread use of telehealth. The authors have adequately addressed the comments made by the reviewers in the first round of review, especially pertaining to the statistical analyses and acknowledging the study’s limitations regarding the small sample size, the short-term testing of the intervention and outcomes and the quasi-experimental design.

Response: The authors thank this reviewer for their thoughtful and positive review of our manuscript.

Comment 2: Authors need to add to the discussion that longer term studies are needed, not only to test the long-term effects of LC in type 2 diabetes [correction: type 1 diabetes], but also to test the LC diet sustainability and its subsequent effects on improved caloric intake and weight loss, which could be mediating factors for improved health outcomes. This is especially important, given that LC diets are known to induce weight loss in the short term only, and that their effects on weight loses its significance in the longer term, compared to other diets.

Response: The authors agree that additional clinical trials are needed to better understand the long-term effects of LC diets on weight management for T1D, as well as to understand the effect(s) of LC diets on T1D management outcomes during energy balance, given that weight loss has been associated with improved health outcomes, including glycaemic control. 

The relevant paragraph of the discussion (page 24, lines 458-462) has been updated to include the following: 

"Future studies should examine the long-term effects (>12 weeks) of LC diets on weight management and diet sustainability in T1D. Further, randomised controlled trials which aim to control for the potential effect(s) of weight loss on glycaemic control outcomes under energy balance conditions are also worthy of consideration."

Comment 3: Additionally, authors need to stress in the discussion on the importance of diet quality when recommending a LC diet for type 1 diabetes management, like emphasizing whole foods and healthy protein and fat sources rather than processed sources, as described in the study protocol (Ref 30: Turton JL, Brinkworth GD, Parker HM, Lee K, Lim D, Rush A, et al. Effects of a low-carbohydrate diet in adults with type 1 diabetes: an interventional study protocol. 2021.8(3):11. Epub 2021-07-22. doi: 10.18203/2349-3259.ijct20212846.), which might also be another contributing factor for the favorable health outcomes.

Response: The authors agree with the need to acknowledge the potential effect(s) of improving diet quality on T1D management outcomes in this study. As such, a paragraph discussing this has now been added to the discussion.

The relevant paragraph of the discussion (page 27, lines 541-552) reads as:

"In the present study, the LC diet intervention prescribed the consumption of minimally-processed whole foods and to minimise the intake of ultra-processed foods. However, due to the preliminary nature of this single arm study that did not assess diet quality, it is difficult to determine the specific effect(s) of changes in diet quality on the favourable metabolic changes observed. Prioritisation of minimally-processed foods has been identified as a core feature of safe and effective LC diet interventions used in T2D management.(32) In addition, common health-promoting dietary patterns, such as the Mediterranean diet, also prioritise consumption of minimally-processed foods such as dairy, nuts, seeds, legumes, meat, fish, and eggs, while limiting ultra-processed foods.(92,93) Future randomised controlled studies should aim to control for the potential impact(s) of diet quality on T1D management outcomes when comparing LC diets with diets higher in carbohydrates. Nevertheless, it would be prudent to consider diet quality in the design and delivery of LC diets in clinical practice."

---

## [Editor Report · Decision Letter 2]

28 Jun 2023

Effects of a low-carbohydrate diet in adults with type 1 diabetes management: a single arm non-randomised clinical trial

PONE-D-22-32366R2

Dear Jessica Turton,

We’re pleased to inform you that your manuscript has been judged scientifically suitable for publication and will be formally accepted for publication once it meets all outstanding technical requirements.

Kind regards,

Jennifer Annette Campbell, PhD, MPH

Academic Editor

PLOS ONE
---

## [Editor Report · Acceptance letter]

3 Jul 2023

PONE-D-22-32366R2 

Effects of a low-carbohydrate diet in adults with type 1 diabetes management: a single arm non-randomised clinical trial 

Dear Dr. Turton:

I'm pleased to inform you that your manuscript has been deemed suitable for publication in PLOS ONE. Congratulations! Your manuscript is now with our production department. 

Kind regards, 

on behalf of

Dr. Jennifer Annette Campbell 

Academic Editor

PLOS ONE